# Bacteria, Fungi and Microalgae for the Bioremediation of Marine Sediments Contaminated by Petroleum Hydrocarbons in the Omics Era

**DOI:** 10.3390/microorganisms9081695

**Published:** 2021-08-10

**Authors:** Filippo Dell’ Anno, Eugenio Rastelli, Clementina Sansone, Christophe Brunet, Adrianna Ianora, Antonio Dell’ Anno

**Affiliations:** 1Department of Marine Biotechnology, Stazione Zoologica “Anton Dohrn”, Villa Comunale, 80121 Naples, Italy; clementina.sansone@szn.it (C.S.); brunet@szn.it (C.B.); ianora@szn.it (A.I.); 2Department of Marine Biotechnology, Stazione Zoologica “Anton Dohrn”, Fano Marine Centre, Viale Adriatico 1-N, 61032 Fano, Italy; eugenio.rastelli@szn.it; 3Department of Life and Environmental Sciences, Polytechnic University of Marche, Via Brecce Bianche, 60131 Ancona, Italy

**Keywords:** marine sediments, petroleum hydrocarbons, hydrocarbonoclastic bacteria, fungi, microalgae, biostimulation, bioaugmentation, genome mining

## Abstract

Petroleum hydrocarbons (PHCs) are one of the most widespread and heterogeneous organic contaminants affecting marine ecosystems. The contamination of marine sediments or coastal areas by PHCs represents a major threat for the ecosystem and human health, calling for urgent, effective, and sustainable remediation solutions. Aside from some physical and chemical treatments that have been established over the years for marine sediment reclamation, bioremediation approaches based on the use of microorganisms are gaining increasing attention for their eco-compatibility, and lower costs. In this work, we review current knowledge concerning the bioremediation of PHCs in marine systems, presenting a synthesis of the most effective microbial taxa (i.e., bacteria, fungi, and microalgae) identified so far for hydrocarbon removal. We also discuss the challenges offered by innovative molecular approaches for the design of effective reclamation strategies based on these three microbial components of marine sediments contaminated by hydrocarbons.

## 1. Introduction

Different chemical contaminants, such as heavy metals and metalloids, aliphatic and polycyclic aromatic hydrocarbons, and halogenated compounds, are introduced in the marine environment by multiple sources (e.g., improper industrial discharges, waste disposal practices, combustion, continental runoff [1,2]). Such pollutants represent a serious risk for marine biodiversity as well as the provision of ecosystem goods and services for human wellbeing [3]. Negative impacts are even more accentuated in coastal ecosystems, characterized by high anthropic pressure and reduced hydrodynamic regime, which may lead to a high accumulation of chemical contaminants in the sediment [4].

Petroleum hydrocarbons (PHCs) contaminating aquatic environments are mainly composed of three classes of compounds: alkanes, olefins, and aromatics [5].

The PHCs, due to poor solubility in water, are adsorbed by particulate matter showing a long-term persistence on the bottom of sediments with a significant negative impact on benthic aquatic communities [6,7].

Different physicochemical techniques have been developed to reduce pollutant concentrations from water (e.g., based on chemical precipitation, ion-exchange, reverse osmosis, electro-dialysis, and ultrafiltration) [8] or for ex-situ treatments of contaminated sediments (e.g., based on chemical, electrochemical, and thermal strategies) [9,10]. However, there is an urgent need to find sustainable and eco-compatible solutions for the remediation of contaminated sediments in situ. Accordingly, international policies (WFD 2000/60 EU [11]) are increasingly seeking management alternatives able to reduce sediment handling interventions, especially by promoting eco-compatible technologies for the decontamination of polluted matrices. Bioremediation, an environmental-friendly and low-cost strategy [12], relies on the ability of microorganisms (including prokaryotes, fungi and microalgae) to reduce contaminant concentrations and/or their toxicity (Figure 1) [13].

Common microbial-based bioremediation strategies include the addition of specific compounds to stimulate autochthonous microbial assemblages (biostimulation) and/or the addition of specific microbial taxa, which display useful biodegradation/detoxification capacity (bioaugmentation [14,15,16,17,18]).

Microbial taxa potentially beneficial for the bioremediation of contaminated sediments can be originally from the same area, or can be isolated from other contaminated areas [19,20]. The use of autochthonous microorganisms is expected to be more effective and ecologically friendly, since these are likely better adapted to the specific local environmental conditions than allochthonous microbes, which may require the manipulation of the natural environment to maximize their performance (e.g., changing oxygen and/or nutrient concentration, pH [21]).

Among bacteria involved in bioremediation processes, most belong to the genera Alcaligens, Achromobacter, Acinetobacter, Alteromonas, Arthrobacter, Burkholderia, Bacillus, Enterobacter, Flavobacterium, Pseudomonas [22,23]. Moreover, genera such as Alcanivorax, Marinobacter, Thallassolituus, Cycloclasticus, and Oleispira include hydrocarbonoclastic bacteria (OHCB) and are known for their specific ability to degrade hydrocarbons [24].

Besides bacteria, fungi have been described for their ability to produce particular enzymes (e.g., catalases, peroxidases, laccases) able to degrade organic contaminants and/or to immobilize inorganic contaminants [25,26]. Fungi belonging to the genera *Aspergillus*, *Curvularia*, *Drechslera*, *Fusarium*, *Lasiodiplodia*, *Mucor*, *Penicillium*, *Rhizopus*, *Trichoderma* were reported as able to degrade aromatic hydrocarbons [27,28,29].

Microalgae, mainly green algae, belonging to the genera *Selenastrum*, *Scenedemus*, or *Chlorella*, have been demonstrated to be effective in the degradation of polycyclic aromatic hydrocarbons, such as naphthalene, phenanthrene, and pyrene [30,31,32,33] and in the immobilization of metals. The mechanisms enabling microalgae to remove toxic compounds, thus reducing their bioavailability and toxicity, mainly rely on the production of exopolysaccharides, which can mediate the uptake of contaminants on the cell surface and/or their complexation into less bioavailable forms [34,35,36]. The contaminant attached to the membrane or cell wall exopolysaccharides (depending on the microalgal taxa) can remain adherent or internalized and chelated by molecules belonging to the phytochelatin classes [37,38,39].

Overall, current knowledge suggests that the use of microalgal strains, coupled with the use of bacteria and fungi, could be a promising bioremediation strategy for the reclamation of marine environments contaminated by petroleum hydrocarbons.

In this study, we review the current knowledge on the main microbial taxa so far involved in bioremediation, also considering the main factors influencing the bioremediation performance of marine sediments contaminated by petroleum hydrocarbons. Then, we discuss the suitability of innovative molecular approaches, which can be useful to enhance the reclamation efficiency of contaminated marine sediments by hydrocarbons.

## 2. Factors Affecting the Bioremediation of Sediments Polluted by Petroleum Hydrocarbons

The pollution of marine sediments by petroleum hydrocarbons is a widespread problem affecting the coasts of many regions of the globe and represents a major concern for the potential detrimental consequences on ecosystem biodiversity, functioning, and overall health [40,41]. PAHs are considered particularly dangerous to ecosystems, since they are potentially mutagenic and carcinogenic [42,43].

In situ and laboratory studies reported that the biodegradation efficiency of petroleum-contaminated sediments can be increased by enhancing biomass and/or activity of hydrocarbon-degrading microorganisms through biostimulation as well as bioaugmentation strategies [44]. Understanding the factors influencing microbial metabolism and hydrocarbon degradation is crucial to the design of an optimal bioremediation strategy [45,46,47].

The physical nature of the crude oil, including available surface area and number of carbon atoms composing the hydrocarbon chains, is one of the key factors affecting hydrocarbon bioremediation [48,49]. For instance, a single large oil slick has a smaller surface area for oil-degrading microbes to access compared to numerous small-sized oil slicks [50]. Moreover, the chemical nature of the spilled petroleum plays a key role in biodegradability. Heavy molecular weight hydrocarbon compounds can be more recalcitrant than lighter ones, which are easier for microbes to be metabolized due to their higher rate of diffusion through the oil-water interface [51]. In addition, unbranched alkanes can be degraded more easily than branched alkanes or multiple-ringed aromatic hydrocarbons [49,52].

The degradation rate of hydrocarbons is also influenced by the availability of nutrients as well as by environmental conditions [53,54,55,56]. Nitrogen and phosphorus have been identified as the two most limiting factors for bacterial-mediated hydrocarbon degradation [57,58], but even sulfur and potassium availability can affect bioremediation rates [58]. Crude oil degradation is faster in warm water as heat promotes the breakdown of the spilled petroleum that becomes more available to oil-degrading microbes [59]. Conversely, sub-zero temperatures cause the shutdown of transport channels of cells and slow down cytoplasm flow processes, hampering or inactivating microbial metabolism and hence their biodegradation potential. Moreover, despite some microbes being cold-tolerant, freeze-thaw seasonal cycles between winter and summer may limit the bioavailability of the spilled petroleum, thus contributing to reduce microbial biodegradation efficiency [60].

Sediment grain size can also affect bioremediation yield by influencing the availability of hydrocarbons from microbial attack and the diffusion of nutrients and dissolved gases needed to support microbial metabolism [53]. Adsorption mechanisms on the small sediment particles such as silt/clay can reduce microbe–hydrocarbon interactions, while the reduced interstitial spaces of silt-clay-dominated sediments can limit gas and solute exchanges and thus their bioavailability for microbial activity [61]. Moreover, pH, by influencing microbial metabolism, can play a role in affecting bioremediation performance, which typically is effective at pH values around 6–8 [62].

Oxygen concentration is another factor influencing bioremediation processes, as most petroleum-degrading microbes identified so far are aerobic (Table 1).

Marine benthic systems characterized by reduced oxygen availability (e.g., sediments of the oxygen minimum zones, sediments characterized by high organic matter content in highly eutrophic systems, or sub-surface sediments) display lower biodegradation rates of hydrocarbon compared to fully oxygenated systems [15]. Indeed, aerobic conditions favor PAHs degradation through oxygenase-mediated activities [33]. Usually, the hydroxylation of an aromatic ring via a dioxygenase, a multi-component enzyme consisting of reductase, ferredoxin, and terminal oxygenase subunits, represents the first step in the aerobic bacterial degradation of PAHs [19]. Dioxygenase activity allows the formation of cis-dihydrodiol, re-aromatized to a diol intermediate via dehydrogenase. Intra diol or extra diol ring-cleaving dioxygenases, through either an ortho-cleavage or meta-cleavage pathway, cleave diol intermediates, promoting the formation of catechols or protocetechuate. Such intermediates are then converted, through β-ketoadipate pathway, to citric acid cycle (CAC) intermediates [88]. Gentisate, homogentisate, and homoprotocatechuate represent other metabolic routes, whose genes have been described in metagenomes and trascriptomes belonging to *Pseudomonas aeruginosa* PAO1, *Klesbiella Pneumoniae* AWD5, and within a bacteria consortium composed by *Pseudomonas*, *Aquabacterium*, *Chryseobacterium*, *Sphingobium*, *Novosphingobium*, *Dokdonella*, *Parvibaculum,* and *Achromobacter* [89,90,91]. The cytochrome P450-mediated pathway is a further metabolic pathway used by bacteria to degrade PAHs, which leads to the production of trans-dihydrodiols [92].

PAHs breakdown also occurs under anaerobic conditions, e.g., under nitrate/sulfate reducing conditions [93], with a great body of literature indicating that bioremediation can be effective also in anoxic conditions [94,95]. In marine anoxic environments, the reductions of sulfate, Mn(IV), and Fe(III) represents the primary source for terminal electron-accepting processes [96] (Figure 2).

Thus, the breakdown of hydrocarbons mediated by microbial metabolism under anaerobic conditions can be successful if the hydrocarbon oxidizers are sulfate, Fe(III), or Mn(IV) reducers [97]. Previous studies demonstrated that hydrocarbon degradation coupled with sulfate reduction prevails in marine anoxic sediments, since sulfate is usually more available than Fe(III) [97]. The biostimulation of sulfate reducers is thought to be a suitable strategy for promoting the biodegradation of hydrocarbons in anoxic marine sediments [98]. Even though different bacterial strains have been identified as capable to degrade a large variety of petroleum contaminants in anoxic marine sediments, strategies and tools able to increase microbial growth and biodegradation performance still need to be investigated and optimized.

## 3. Bacterial-Mediated Degradation of Petroleum Hydrocarbons

The widespread contamination of marine systems by hydrocarbons, especially in coastal areas characterized by high anthropogenic pressure, has stimulated research focused on the identification of the bacterial taxa most effective in their removal [99]. The bacteria most often associated with the presence of oil in the sea include those belonging to the gammaproteobacteria, especially members of *Oceanospirillales* and *Alteromonadales* [20,24], including hydrocarbon degraders (Table 2), such as *Alcanivorax*, *Halomonas*, *Marinobacter*, *Oleispira*, *Thalassolituus*, and *Oleiphilus*.

The high degradation yields of bacteria in marine environments are partially due to their halophilic features allowing them to operate under different levels of saline stress [116]. Ref. [117] reported the ability of an halophilic consortium composed by *Halomonas*, *Dietzia*, and *Arthrobacter* to degrade about 40% of diesel oil. *Halomonas* strains seem to be highly suitable for hydrocarbon degradation ability. Indeed, a *Halomonas* strain SZN1 [19] displayed a degradation rate between 47% and 80% when incubated with sediments contaminated by pyrene, indeno pyrene, chrysene, and dibenzo anthracene.

Even though hydrocarbon degrading halophilic or halotolerant bacteria have been mainly associated with *Marinobcater*, *Alcanivorax*, *Halomonas,* and *Dietzia* [118], new studies are highlighting the possibility of employing strains less known for bioremediation purposes. Indeed, halophilic bacteria consortia involving *Ochrobactrum halosaudis*, *Stenotrophomonas maltophilia*, *Achromobacter xylosoxidans*, and *Mesorhizobium halosaudis*, are effective for the degradation of phenanthrene, fluorene, and pyrene [119]. Similarly, another marine bacterium *Halorientalis* sp. has been described as capable to degrade hexadecane with a degradation rate of 57% at 3.6 M NaCl [120].

Recently, [121] reported the ability of halophilic *Staphylococcus* CO100, isolated from Tunisian contaminated marine sediments, to successfully degrade 72% of the aliphatic hydrocarbons contained in crude oil (1%, *v*/*v*) after 20 days of culture at 100 g/L NaCl. Another example of a marine bacterium effective in the removal of oil from contaminated systems is represented by the biosurfactants’ producer *Paracoccus* sp. MJ9, capable of removing up to 80% of diesel oil in about five days [122].

The use of surfactant compounds produced by microorganisms represents a promising approach to improve the bioremediation efficiency of polluted environments [123]. Such compounds, thanks to their amphiphilic moieties, favor bioremediation processes by promoting the partitioning of the hydrophobic contaminants into internal hydrophobic cores of surfactant micelles, which ultimately facilitates the detachment of pollutants from the sediments [124]. Indeed, one of the factors limiting the biodegradation of hydrocarbons is their poor bioavailability due to their hydrophobic nature, and biosurfactants can help to increase the bioavailability of hydrocarbons for microbial cells. The effectiveness of a particular class of biosurfactants, rhamnolipids, in the remediation of marine crude oil contaminated matrixes have been shown by [125]. The addition of rhamnolipids to a solution of crude oil and sand has led, after 15 days, to a degradation yield of 30% for fluorene, ca. 20% for phenanthrene, and 10% for dibenzothiophene. Recently, it has been reported that the addition of a rhamnolipid like biosurfactant, produced by a halotolerant *Pseudomonas aeruginosa* (AHV-KH10), can allow a biodegradation yield of diesel up to 70% [126]. High removal rates (up to 80%) of total petroleum hydrocarbons from contaminated matrices have been also reported by adding a mixture of rhamnolipids, biochar and nitrogen [127]. This result suggests that a combined use of biosurfactants and compounds capable of stimulating the metabolism of the autochthonous microbial taxa may be an effective solution to increase bioremediation processes.

A novel approach to speed up the bioremediation of oil spills in marine ecosystems is represented by the use of new materials as possible carriers of hydrocarbonoclastic bacteria. For instance, Ref. [128] reported the possibility of using a carrier constituted by puffed *Panicum miliaceum* (PPM), calcium alginate and chitosan, able to immobilize degrading oil microorganisms in its porous structure. This structure has been shown to be biodegradable and float on the oil-contaminated seawater. Moreover, to solve the problem of nutrient supply in highly contaminated seawater, the authors added to the carrier structure an emulsion composed by urea solution, soybean lecithin, alcohol, and oleic acid to be used as oleophilic fertilizer. Field mesocosm experiments conducted in coastal seawater showed that most of the petroleum hydrocarbons (>98%) were removed from the surface of seawater in 24 h [128]. Similarly, the addition of solid inoculants of freeze dried *Bacillus subtilis* LZ-2 bacteria enhanced the degradation rate of crude oil by 44.2% and 21.6% for total saturate and aromatic hydrocarbons, respectively [129]. These results highlight the potential of these approaches for bioremediation of hydrocarbons contained within the crude oil. However, when designing bioremediation strategies for the removal of hydrocarbons, contaminants co-occurring in the environmental matrix should also be taken into account [130]. Indeed, biodegradation strategies could induce important changes in the mobility and bioavailability of heavy metals, possibly increasing environmental risk [19,98,131,132]. Therefore, an accurate risk analysis should be conducted to assess the contextual effects of the biotreatments, especially for sediments characterized by mixed chemical contamination (i.e., organic and inorganic pollutants).

## 4. Fungi-Mediated Degradation of Petroleum Hydrocarbons

Theoretical estimates suggest that fungi can be the most diversified group of unicellular eukaryotes on Earth with more than 5 million species, of which only 5% have been described [133,134]. Thus, marine ecosystems could represent a reservoir of still uncharacterized fungi with promising biotechnological applications [135].

Although research has been mostly focused on bacteria for bioremediation purposes, the use of fungi (i.e., mycoremediation) has gained increasing attention due to the ability of such microorganisms to produce a different array of enzymes capable of transforming a wide variety of hazardous chemicals [18,34,136]. Fungi, besides releasing extracellular enzymes in the surrounding environments, may display a strong degrading activity due to their hyphal network [137]. Indeed, the typical apical growth of fungi allows them to penetrate (contaminated) sediments more easily than others microorganisms [138]. Additionally, thanks to a wide metabolic plasticity, fungi can assimilate hydrocarbons in environments characterized by low nutrient concentrations and low pH [139].

Different genera of fungi, belonging to the *Cladosporium*, *Aspergillus*, *Cunninghamella*, *Penicillium*, *Fusarium*, and *Mucor* have been described to be involved in aliphatic hydrocarbon degradation, as well as in the breakdown of more recalcitrant aromatic hydrocarbons [135]. Fungal species, such as *Phanerochaete chrysosporium*, *Agaricus bisporus*, *Trametes versicolor*, and *Pleurotus ostreatus*, are effective for the decontamination of polluted sites [140].

Although several terrestrial fungal taxa have been described as playing a prominent role in hydrocarbon biodegradation, to date, only a few studies have investigated fungal bioremediation potential of petroleum hydrocarbons in marine environments. Nevertheless, different studies have provided evidence that fungi identified in different marine ecosystems can be effective in the degradation of petroleum hydrocarbons [141,142,143,144] (Table 3).

An example of an effective treatment based on mycoremediaton has been reported by [145], who identified, among isolated fungal taxa (belonging to *Aspergillus*, *Fusarium*, *Penicillum*, and *Acremonium*) from different marine sediments, the fungal species *Penicillium citrinum,* capable of reducing the mass of the total crude oil by 77% and that of the individual n-alkane on average by 95%. Similarly, Ref. [146], collecting hydrocarbon-contaminated sediments from the west coast of Algeria, reported the ability of 12 fungal strains belonging to the genera *Penicillium*, *Aspergillus* and *Cladosporium* to efficiently degrade crude oil as well as to produce biosurfactants.

Also fungi isolated from deep-sea oil reserves in the gulf of Mexico have revealed promising biotechnological applications for the reclamation of hydrocarbon contaminated systems [147]. Indeed, the isolated strains comprising *Aureobasidium* sp., *Penicillium brevicompactum*, *Penicillium* sp., *Phialocephala* sp., and *Cladosporium* sp. 1, 2, and 3 were characterized by the ability to degrade and grow on hexadecane and 1-hexadecene as the sole carbon sources. Indeed, gene expression analysis revealed the upregulation of transmembrane transporters (genes HXT3, RAG1, and GHT6), suggesting the incorporation of the tested hydrocarbons within the cell and pointing towards a possible application in hydrocarbon bioremediation.

Fungi can also degrade PAHs. Indeed, *Aspergillus sclerotiorum* CBMAI 849 and *Mucor racemosus* CBMAI 847, fungi isolated from the northern coast of Brazil, were able to metabolize between 50% and 90% of pyrene and benzo[a]pyrene [148]. Aromatic compounds have been also reported to be efficiently degraded by three marine basidiomycetes: *Tinctoporellus* sp. CBMAI1061, *Peniophora* sp. CBMAI106 and *Marasmiellus* sp. CBMAI 1062 [149]. In particular, the latter was able to degrade almost 100% of pyrene (0.08 mg mL^−1^) after 48 h of incubation. The analysis of intermediate metabolites of pyrene degradation contributed towards unveiling molecular mechanisms driving hydrocarbon degradation, suggesting that the mycoremediation of aromatic compounds could be carried out through the cytochrome P450 system and epoxide hydrolases.

Fungi metabolic pathways involved in hydrocarbon breakdown in aerobic conditions may follow an initial oxidation mediated by cytochrome P450 monooxygenases and alkane-oxygenase enzymes [150]. The degradation of PAHs by ligninolytic fungi have been extensively studied in the past few years [151], especially in white rot fungi. These fungi produce extracellular peroxidases which are responsible for the initial oxidation of PAHs [152]. The principal mechanism of biodegradation used by white rot fungi relies on the lignin degradation system which involves extra-cellular lignin modifying enzymes (LMEs), displaying reduced substrate-specificity. LMEs lead to the mineralization of a wide range of highly recalcitrant organic pollutants that are structurally similar to lignin. The major components of the lignin degradation system include lignin-peroxidase (LiP), manganese peroxidase (MnP), H_2_O_2_ producing enzymes, and laccase, although not all ligninolytic fungi display the complete set of enzymatic activities [153].

Thanks to increased knowledge on the metabolic pathways expressed in microorganisms and to technological advances in synthetic biology, new horizons have been opened in the field of bioremediation. To this aim, the characterization of possible biosynthetic fungal clusters associated with laccase expression is promising, e.g., to enhance their expression in heterologous yeast hosts [137]. Indeed, fungal enzymes, ad-hoc targeted for the degradation of specific pollutants, could be applied to contaminated sites for biodegradation purposes. However, the use of such an approach is still in its infancy and needs to be optimized due to high costs, low yield, and the difficulty of obtaining stable products once released on the polluted matrix.

## 5. Microalgae Involved in Hydrocarbon Removal

A possible alternative to hydrocarbon remediation mediated by bacteria or fungi is represented by phytoremediation, which consists in the use of plants or algae for the removal of environmental pollutants or their transformation into less harmful substances [22]. In particular, photosynthetic unicellular prokaryotes or eukaryotes, such as cyanobacteria, green, brown, and red algae, represent promising candidates for bioremediation applications due to their high growth efficiency and biomass production [154].

The possibility of exploiting the activity of microalgae for the degradation of aromatic compounds such as naphthalene has been reported almost half a century ago [155]. More recently, [156] reported a pyrene degradation ranging from 34% to 100% during seven days of treatment using the green microalgae (*Chlamydomonas*, *Chlorella*, *Scenedesmus*, *Selenastrum*) or *cyanophyte* (*Synechocystis*). Similarly, other studies revealed that *Skeletonema costatum* and *Nitschia* sp. were effective in the removal of phenantrene and fluoranthrene [157].

In addition, the green microalga *Chlorella vulgaris* displayed a high potential in the remediation of waters contaminated by crude oil, with a bioremediation efficiency between 88% and 94% [73]. Ref. [158] identified *Chlorella vulgaris* BS1 as capable of degrading 98% of petroleum hydrocarbons at initial concentrations of 115 mg L^−1^ from waters in 14 days. [159] reported that five cyanophytes, namely *Westiellopsis prolific*, *Anabaena variabilia*, *Oscillatoria pranceps*, *Phormidium mucicola*, and *Lyngbya digueti*, were capable of reducing the concentrations of different hydrocarbon compounds from oil refinery waste waters by between 24% and 92%.

Although many microalgal species able to remove hydrocarbons from contaminated environments have been identified from freshwater ecosystems, marine ecosystem can also host effective phototrophic organisms useful for hydrocarbon decontamination. Moreover, the use of marine photosynthetic organisms has several advantages over their freshwater counterparts, such as the use of saltwater instead of freshwater for cultivation and the ability to convert solar energy up to four times more efficiently [160].

To this aim, marine cyanobacteria *A. quadruplicatum*, *Microcoleus Chthonoplastes*, and *Phormidium corium* have been reported as capable of removing phenanthrene [161]. Another marine taxon belonging to the genus *Phormidium*, isolated from microalgal mats of coastal environments of Todoa Santos Bay (Mexico), was effective for hydrocarbon removal, being able to remove about 45% and 37% of hexadecane and diesel oil from seawater within 10 days, respectively [162]. Moreover, *Nannochloropsis oculata* (eustitgmatophyte) and *Isochrysis galbana* (haptophyte) have been described as promising candidates for the removal of hydrocarbons from contaminated seawater (removal yield of about 80% [163]).

Although the ability of microalgae and/or cyanobacteria to interact with petroleum compounds is established, the biochemical and physiological mechanisms enabling the removal of hydrocarbons need to be better clarified. Indeed, the model based on biosorption and bioaccumulation, despite being useful to explain the dynamics of metal removal, is actually not fully applicable to elucidate the mechanisms of hydrocarbon degradation [118,164]. A possible explanation may arise from the observation that cultures of marine phototrophs are usually not axenic, and are colonized by a diversified assemblage of heterotrophic microbes, which could be responsible for the actual degradation of hydrocarbons [164,165,166,167]. In support of this hypothesis, Ref. [71] demonstrated the predominance of hydrocarbonoclastic bacteria (e.g., *Alcanivorax* or *Marinobacter* spp.) following the incubation of two marine microalgae (*Pavlova lutheri* and *Nannochloropsis oculata*) with crude oil.

The synergy of a mixed algae-bacteria system might be related to oxygen produced by the algal photosynthetic process that could foster bacteria to oxidize the pollutants [158] and thus might allow the remove of hydrocarbons. Such “supplementary” oxygen would reduce possible oxygen limitation problems during heterotrophic hydrocarbon degradation, typically affecting bacteria-mediated remediation processes [159]. Bacteria-microalgae synergy might also involve the production of exudates by microalgae that can support bacterial growth, possibly accelerating their oil-degrading activity [118]. On the other hand, microalgae could benefit from bacterial-mediated increase of the bioavailability of trace elements, nutrients, and growth-promoting factors [118,168,169]. For instance, as shown for a freshwater consortium of algae and bacteria [170], pyrene-degrading bacteria could both enhance microalgal growth (through the supply of phytohormones) and be stimulated by microalgal activity, in turn accelerating hydrocarbon degradation.

Marine microalgae–bacteria interactions remain to be further investigated to clarify the processes involved in hydrocarbon degradation [171,172], as well as their actual potential to enhance bioremediation yields in bio-based approaches for the reclamation of contaminated marine sediments. The application of microalgae for marine sediment remediation poses several caveats and requests proper optimization associated with light requirements, hampering the utilization of algae below the sediment surface, or at excessive water depths. An alternative might be the use of mixotrophic or heterotrophic microalgae, but this still needs to be investigated. Further, the development of formulated enzymes and/or products from microalgae, rather than live cells, will be a novel option for marine sediment bioremediation, as proposed for soil remediation [173]. For instance, Ref. [174] have recently shown the possibility to remediate marine sediments contaminated by phenolic chemicals using an innovative biochar derived from a red algae (*Agardhiella subulata*) able to generate reactive radicals under alkaline pH conditions. Moreover, ex situ approaches might be attempted, as recently performed in testing pyrene degradation by *Chlorella* in soil slurry [175].

## 6. Towards an Omics Bioremediation Approach

Our ability to identify microbes capable of degrading contaminants or reducing their toxic effects has been boosted by the development of next generation sequencing (NGS) techniques and in silico analyses [176]. Technologies based on high-throughput analysis are very useful in shedding light on microbial community diversity, otherwise not accessible using culture dependent methods. In this context, analyses based on metagenomics, metatranscriptomics, metaproteomics, metabolomics, and fluxomics techniques, supported by specific bioinformatics pipelines, are providing important information to unveil microbial metabolism and interactions among microbes, that can influence contaminants’ degradation/detoxification pathways [177] and provide a deeper understanding of mechanisms underlying bioremediation processes, as well as of bacterial metabolic processes [178].

Metagenomics has set a milestone in microbiology, allowing for the concurrent computation of thousands of microbial genomes and enabling high-throughput investigations of uncultured organisms [179]. This tool represents a promising strategy for bioremediation purposes, since many public databases now contain a rich pool of genetic sequences for the manipulation and engineering of microbial strains for targeted use in bioremediation efforts [180].

Metagenomics approaches can include sequence-based and/or function-based strategies [181]. Sequence-based metagenomics rely on DNA sequencing from environmental samples for gene identification and microbial genome assembly, and for the identification of metabolic pathways of interest, coupled with microbial taxonomic profiling typically based on 16S or 18S rRNA and/or ITS analysis [182]. Conversely, function-based metagenomics typically targets specific function(s) aiming at identifying the presence of proteins/enzymes involved in a specific pollutant metabolism in the investigated matrix [180]. Function-based metagenomics involves DNA extraction from the environment and, after preliminary analysis to check for the presence of enzymes of interest, DNA fragments can be used to obtain clone libraries using the most suitable hosts, to test the effective enzymatic activities [183].

Metagenomics has been demonstrated suitable for bioremediation purposes since it allows to identify the principal taxa composition in contaminated environments and changes that occur when these taxa are exposed to different forms of pollutants [184,185]. To this extent, [186] reported a shift that occurred within the marine microbial assemblages following an oil spill in the west Antarctica Peninsula. Using a metagenomic approach, the authors documented an enrichment of the bacterial community related to *Actinobacteria* and *Polaromonas naphtalenivorans* followed, one year later, by the disappearance of contamination from surface sediments.

Metagenomics, together with other tools (transcriptomics, proteomics, and metabolomics) can lead to the identification of genes and pathways involved in the biodegradation of different recalcitrant pollutants. The possibility of understanding the degradation potential of each microorganism lays the foundations for the engineering of synthetic microbial consortia optimized for the complete breakdown of specific pollutants, which often require multiple enzymes, hardly possessed by a single microbial strain [187,188].

For the purposes of the present review, we conducted a search in public databases to quantify the occurrence of genes potentially involved in the degradation of hydrocarbons in genomes of currently known hydrocarbonoclastic bacterial genera (*Alcanivorax*, *Cycloclasticus*, *Marinobacter*, *Oleiphilus*, *Oleispira*, and *Thalassolitus*), whose degradation capacity has been largely documented in the literature (Table 4).

We included in this search 17 bacterial genomes and 20 genes involved in both the main and the peripheral degradation pathways of hydrocarbons, such as alkane hydroxylase, naphtalene dioxygenase, cytochrome P450, catechol dioxygenase, and protocatechuate 3,4-dioxygenase [189].

The most represented genes involved in hydrocarbon degradation were cyclohexanol dehydrogenase (representing on average 33% of the total hits in each genome), salicylaldehyde dehydrogenase (17%), alkane hydroxylase (11%) cyclohexanone monooxygenase (10%), followed by naphthalene dioxygenase (8%) ferrodoxin reductase (6%), alcohol dehydrogenase (5%), ethylbenzene dioxygenase (4%), and cytochrome P450 (2%) (Figure 3).

All the other genes known to be involved in petroleum hydrocarbon degradation showed a lower frequency or were absent in the analyzed genomes. The genome mining analysis also highlighted that six of the twenty identified genes (i.e., alkane hydroxylase, alcohol dehydrogenase, cyclohexanol dehydrogenase, cyclohexanone monooxygenase, ferrodoxin reductase, and salicylaldehyde dehydrogenase) were present in all of the 17 genomes investigated. This result partly confirms previous findings based on the analysis of bacterial assemblages from oil polluted sediments in the northern Gulf of Mexico [191], which showed large quantities of cyclohexanol dehydrogenase and alcohol dehydrogenase. Even though the presence of still-unknown genes involved in hydrocarbon degradation pathways cannot be ruled out, our results suggest that these six genes, present in all marine hydrocarbonoclastic bacteria investigated, may represent the core functional set in the degradation of alkanes and cyclic hydrocarbons.

Since genomics and metagenomics have limitations related to gene expression and protein activity [192], transcriptomics and metatranscriptomics could help to identify new genes of biotechnological interest. RNA sequencing allows the evaluation of the expression of genes with bioremediation potential and provides an indirect estimate of microbial activity, representing a better target than DNA to assess the degradation ability of a given microbial assemblage towards specific contaminants. For instance, metatranscriptomics has been used to identify up-regulated genes under contamination conditions and to identify potential novel genes involved in bioremediation [192]. Nonetheless, a microarray/transcriptomic based approach, involving the use of short oligomers to determine gene expression after incubation with contaminants, has been used to develop a synthetic microbial consortium with bioremediation potential [193].

Environmental proteomics and metabolomics are promising tools, and are often coupled with transcriptomics [194]. Proteomics and metaproteomics rely on protein extraction (from culture media or environmental samples), followed by a separation phase on acrylamide gel (2D-GE) and identification of the product by mass spectroscopy [195]. Such techniques find their applications in the investigation of proteins expressed by microorganisms under harsh environmental conditions and allow to understand the molecular basis of protein folding [178]. Despite metatranscriptomics representing a valuable approach to assess the physiological changes occurring in microorganisms in response to environmental stimuli, the metaproteomics tool has some advantages, as proteins are more stable than mRNAs, which can otherwise be degraded or translated inefficiently. Thus, metaproteomics can provide a better snapshot of biological mechanisms expressed in situ, since it is likely less affected by extraction procedures compared to transcriptomics [196].

To this extent a proteomic analysis conducted on *Pseudomonas putida* KT2440 incubated with aromatic compounds has allowed the identification of about 110 proteins involved in hydrocarbon degradation pathways, including benzoate dioxygenase (BenA, BenD), catechol 1,2-dioxygenase (CatA), protocatechuate 3,4-dixoygenase (PcaGH), β-Ketoadipyl CoA thiolase (PcaF) and 3-oxoadipate enol-lactone hydrolase (PcaD) [197].

Similarly, the proteomic approach has recently allowed identifying about 250 proteins involved in hydrocarbon degradation pathways in *Pseudomonas* sp. ISTPY2 grown in the presence of pyrene, suggesting phthalate 4,5-dioxygenase, aldehyde dehydrogenase, and F420-dependent oxidoreductase as the main drivers of the bioremediation process [198].

Metabolomics is an omics tool allowing to separate and identify molecules using gas chromatography and mass spectroscopy, respectively [199]. This approach aims to characterize the end product of enzyme activity and differs from proteomics, which otherwise provides information about the total protein pattern expressed [200]. Generally, metabolome-based approaches, including metabolism-based wide fluxes (fluxomes), allow evaluating the effects of toxic substances and the mediated responses of microorganisms. They can also unveil the nature of molecules guiding the complex interactions in consortia degrading pollutants, providing further knowledge on how to optimize bioremediation strategies [177]. An example of the effectiveness of metabolomic techniques in identifying metabolites produced by microorganisms under stress conditions is the nuclear magnetic resonance (NMR)-based metabolomic approach. This method relies on high-throughput fingerprinting analysis capable of unravelling the metabolites of interest for a specific organism without prior knowledge. For this purpose, a novel freshwater microalga *Scenedesmus* sp. IITRIND2, capable of growing at high concentrations of As, was chosen to analyse its metabolic profile under As (III) and As (V) stress conditions. The results showed that about 18 metabolites related to the metabolism of free amino acids, carbohydrates, and ATP were involved in the mitigation mechanisms of the toxicity of As [201]. Similar approaches are thus envisaged for future studies on the microbial-mediated bioremediation of marine sediments contaminated by petroleum hydrocarbons.

Overall, despite the fact that the application of different omics tools is leading to unprecedented knowledge about adaptation mechanisms and the metabolism of microorganisms, a culture dependent approach is still required to identify suitable isolates which can be massively grown and used for bioremediation purposes. Future research should be based on the combination of multiple approaches for improving the current knowledge on processes and interactions that take place in contaminated marine sediments. This is a prerequisite for developing efficient eco-sustainable strategies based on biostimulation or bioaugmentation approaches for the recovery of contaminated marine sediments, in view of large-scale applications.

## 7. Concluding Remarks

Our review highlights that a variety of microbial taxa, belonging to bacteria, fungi, and microalgae, are able to degrade hydrocarbons and are thus potentially useful for the remediation of contaminated marine sediments. However, different microbial taxa have different metabolic requirements and can demonstrate differing efficiency in the biodegradation of petroleum hydrocarbons, which can also greatly vary depending upon the chemical structure and bioavailability of the hydrocarbons and environmental conditions. Future research should be devoted to understanding potential synergistic interactions among microbial taxa and to assess their potential in hydrocarbon removal following in situ and ex situ bioremediation applications. More information is required concerning microbial–hydrocarbon interactions and degradation pathways. This is an important prerequisite for the design of effective and eco-friendly in situ applications, which require a robust assessment of the potential detrimental ecological effects due to incomplete biotransformation, which could determine an increase of the ecotoxicological risks of pollutants present in the sediment. Culture-independent techniques represent a promising approach for the gene mining of otherwise inaccessible marine microorganisms without the requirement of culturing efforts and can facilitate the discovery of novel hydrocarbon-degrading microbial taxa. Finally, the performance of in situ sediment bioremediation has to be tested at large scale, by scaling-up laboratory or small-scale investigations and including appropriate estimates of economic costs and possible environmental impacts. These represent key aspects for the development of sustainable and eco-compatible bioremediation interventions on marine sediments contaminated with petroleum hydrocarbons.

## Figures and Tables

**Figure 1 microorganisms-09-01695-f001:**
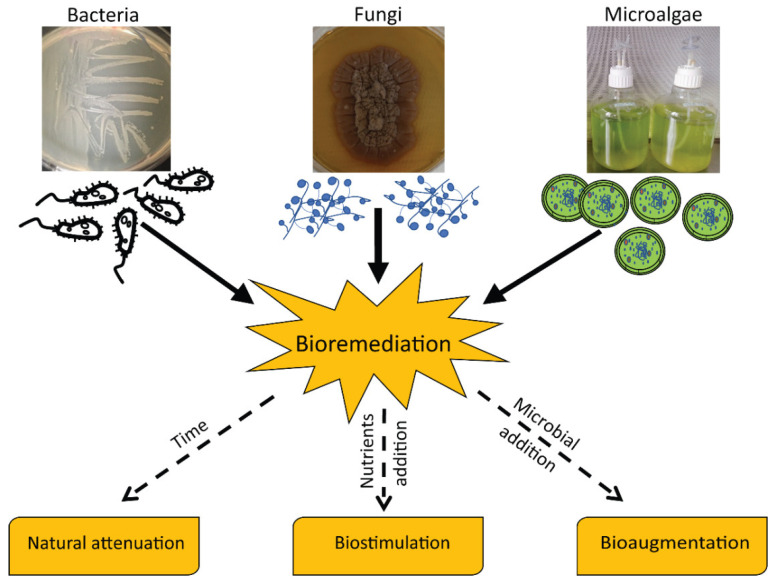
General scheme of bioremediation strategies involving different microbial taxa.

**Figure 2 microorganisms-09-01695-f002:**
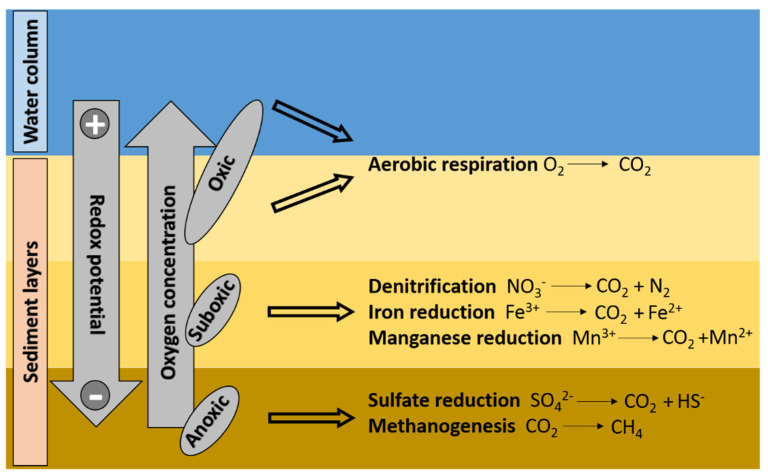
Conceptual scheme of metabolic processes involved in hydrocarbon degradation in marine sediments characterized by different redox conditions.

**Figure 3 microorganisms-09-01695-f003:**
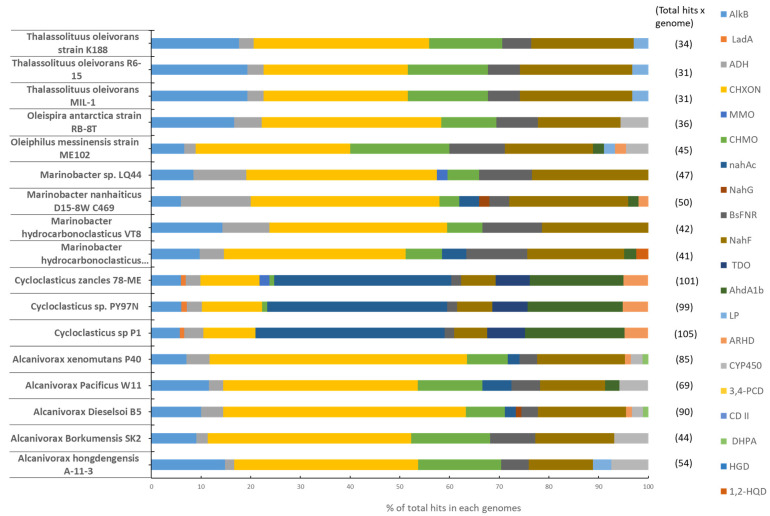
Relative abundance (expressed as % of the total hits) of genes involved in the degradation of aliphatic and aromatic hydrocarbons within the genomes of hydrocarbonoclastic bacteria. For each bacterial genome, the relative abundance of a single query gene has been calculated as the ratio between the number of genomic hits to that specific query gene, and the total number of genomic hits to all the 20 query genes involved in hydrocarbon degradation. The analysis was conducted using the standalone version of BLAST (Bit-Score > 80; e-Value < 0 e-15; [190]) to document the presence of the genes of interest within the selected genomes. The legend on the right refers, from top to bottom, to (AlkB) alkane hydroxylase, (LadA) alkane monooxygenase LadA, (ADH) alcohol dehydrogenase, (CHXON) cyclohexanol dehydrogenase, (MMO) methane monooxygenase, (CHMO) cyclohexanone monooxygenase, (nahAc) naphthalene dioxygenase, (NahG) salicylate 1-monooxygenase, (BsFNR) ferrodoxin reductase, (NahF) salicylaldehyde dehydrogenase, (TDO) toluene dioxygenase, (AhdA1b) ethylbenzene dioxygenase large subunit, (LP) lipase, (ARHD) aromatic ring-hydroxylating dioxygenase subunit alpha, (CYP450) cytochrome P450, (3,4-PCD) protocatechuate 3,4-dioxygenase, (CD II) catechol 1,2 dioxygenase, (DHPA) 3,4-dihydroxyphenylacetate 2,3-dioxygenase, (HGD) homogentisate 1,2-dioxygenase and (1,2- HQD) hydroxyquinol 1,2-dioxygenase.

**Table 1 microorganisms-09-01695-t001:** List of the principal genera of bacteria, microalgae and fungi involved in bioremediation.

Organisms	Genus	Ref.
**Bacteria**	*Alcaligens*	[63]
	*Bacillus*	[64]
	*Enterobacter*	[65]
	*Flavobacterium*	[66]
	*Pseuodmonas*	[67]
	*Alcanivorax*	[24]
	*Thallassolituus*	[68]
	*Cycloclasticus*	[69]
	*Oleispira*	[70]
	*Marinobacter*	[71]
**Microalgae/Cyanobacteria**	*Spirulina*	[72]
	*Chlorella*	[73]
	*Spirogyra*	[74]
	*Scenedesmus*	[75]
	*Oscillatoria*	[76]
	*Chlorococcum*	[77]
	*Synechocystis*	[78]
	*Nannochloropsis*	[71]
	*Selenastrum*	[79]
**Fungi**	*Aspergillus*	[80]
	*Curvularia*	[28]
	*Drechslera*	[81]
	*Fusarium*	[81]
	*Lasiodiplodia*	[82]
	*Mucor*	[83]
	*Penicillium*	[84]
	*Rhizopus*	[85]
	*Trichoderma*	[86]
	*Cryptococcus*	[87]

**Table 2 microorganisms-09-01695-t002:** An overview of oil degrading bacteria isolated from different marine ecosystems and their hydrocarbon specificity.

Oil-Degrading Bacteria.	Hydrocarbon Specificity	Reference
*Alcaligenes aquatilis* BU33N	Crude oil and phenanthrene	[100]
*Alcanivorax* sp. IO_7	Alkane	[101]
*Alcanivorax* sp. 24	Alkanes	[102]
*Cupriavidus metallidurans* CH34	Toluene	[103]
*Cycloclasticus* sp. strain BG-2	Phenanthrene	[104]
*Cycloclasticus* sp. 78-ME	Polycyclic aromatic hydrocarbons	[105]
*Cycloclasticus* sp. strain P1	Naphthalene, phenanthrene, pyrene	[106]
*Halomonas* sp. strain MCTG39a	Hexadecane	[107]
*Halomonas pacifica* strain Cnaph3	Naphthalene	[108]
*Marinobacter hydrocarbonoclasticus* SdK644	Crude oil	[109]
*Oleispira* antarctica RB-8	Aliphatic alkanes	[110]
*Pseudomonas aeruginosa* N6P6	Phenanthrene and pyrene	[111]
*Pseudomonas pseudoalcaligenes* NP103	Phenanthrene and pyrene	[112]
*Pseudomonas* sp. sp48	Phenol, naphtalene, pentadecane	[113]
*Pseudomonas aeruginosa* GOM1	Hexadecane	[114]
*Ralstonia pickettii*	Crude oil	[115]

**Table 3 microorganisms-09-01695-t003:** List of the main marine fungi hydrocarbon degraders.

Species	Location	Compounds	Ref.
*Aspergillus sydowii* NIOSN-SK56C42	Deep sea sediment (Arabian sea)	Crude oil, alkanes	[145]
*Acremonium sclerotigenum NIOSN-M109*	Mangrove sediment (Panaji, Goa)		
*Penicillium citrinum NIOSN-M126*	Mangrove sediment (Panaji, Goa)		
*Aspergillusflavus NIOSN-SK56S22*	Deep sea (Arabian sea)		
*Penicillium polonicum*	Marine sediments (Port of Oran, Algeria)	Crude oil	[146]
*Penicillium cyclopium*			
*Penicillium mononematosum*			
*Penicillium chrysogenum*			
*Aureobasidium* sp.	Deep sea oil reserves (Gulf of Mexico)	hexadecane and 1-hexadecene	[147]
*P. brevicompactum*			
*Phialocephala* sp.			
*Penicillium* sp.			
*Cladosporium* sp. 1, 2			
*C. gracilis*			
*Aspergillus sclerotiorum* CBMAI 849	Coastal Atlantic Ocean (São Paulo, Brazil)	pyrene and benzo[a]pyrene	[148]
*Mucor racemosus* CBMAI 847			
*Tinctoporellus* sp. CBMAI1061	Marine sponges (*Dragmacidonreticulatum* and *Amphimedon viridis)*	PAHs, Pyrene	[149]

**Table 4 microorganisms-09-01695-t004:** List of enzyme and genomes used for sequence alignment. Only 100% complete genomes that are publicly available in NCBI database were included in the search.

Enzymes	EC Num.	Genomes	Access. Num.
Alkane hydroxylase	1.14.15.3	*Alcanivorax hongdengensis* A-11-3	NZAMRJ000001
Alkane monooxygenase LadA	1.14.14.28	*Alcanivorax Borkumensis* SK2	NC008260
Alcohol dehydrogenase	1.1.1.2	*Alcanivorax Dieselsoi* B5	NC018691
Cyclohexanol dehydrogenase	1.1.1.245	*Alcanivorax Pacificus* W11-5	NZCP004387
Methane monooxygenase	1.14.13.25	*Alcanivorax xenomutans* P40	NZCP012331
Cyclohexanone monooxygenase	1.14.13.22	*Cycloclasticus* sp. P1	NC018697
Naphthalene dioxygenase	1.14.12.12	*Cycloclasticus* sp. PY97N	NZCP023664
Salicylate 1-monooxygenase	1.14.13.1	*Cycloclasticus zancles* 78-ME	NC021917
Ferrodoxin reductase	1.18.1.2	*M. hydrocarbonoclasticus* ATCC49840	NC017067
Salicylaldehyde dehydrogenase	1.2.1.65	*Marinobacter hydrocarbonoclasticus* VT8	NC008740
Toluene dioxygenase	1.14.12.11	*Marinobacter nanhaiticus* D15-8W C469	NZAPLQ000001
Ethylbenzene dioxygenaseL-sub.	1.14.12.18	*Marinobacter* sp. LQ44	NZCP014754
Lipase	3.1.1.3	*Oleiphilus messinensis* strain ME102	NZCP021425
Aromatic ring-hydroxylating diox.	1.14.12.3	*Oleispira antarctica* strain RB-8	FO203512
Cytochrome P450	1.14.14.1	*Thalassolituus oleivorans* MIL-1	NC020888
Protocatechuate 3,4-dioxygenase	1.13.11.3	*Thalassolituus oleivorans* R6-15	NZCP006829
Catechol 1,2-dioxygenase	1.13.11.1	*Thalassolituus oleivorans* strain K188	NZ_CP017810.1
3,4-dihydroxyphenylacetate diox.	1.13.11.15		
Homogentisate 1,2-dioxygenase	1.13.11.5		
Hydroxyquinol 1,2-dioxygenase	1.13.11.37		

## Data Availability

The data presented in this study are openly available in Table 4.

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
