# Peer review of "Bacteria, Fungi and Microalgae for the Bioremediation of Marine Sediments Contaminated by Petroleum Hydrocarbons in the Omics Era"

_microorganisms, 2021, doi:10.3390/microorganisms9081695_

Round 1
Reviewer 1 Report
Thank you for the opportunity to review the article entitled Bacteria, fungi and microalgae for the bioremediation of marine sediments contaminated by petroleum hydrocarbons in the omics era. The article is very interesting and will probably be a very valuable literature item. The authors made a complete review of bioremediation of contaminated marine sediments. In this review, they referred to 201 literature references. In my opinion, the article is written very well in terms of the content and I have no comments of this nature. However, the article needs to be improved in terms of editing. First of all, the authors have to correct the way of editing the text - extract paragraphs; arrange unnecessary spaces and interlining between the text; change the presentation of information in the tables according to journal template. Moreover, after reading the article, I find that the Authors should consider two things in it. First of them is to add some more figures or schemes that refer to the quoted references. This comment is due to the fact that there are many pages in this article that are completely filled with text only. This form is not friendly to the potential reader and does not encourage him to read further. Adding more figures can significantly improve the reader's perception of this article. My second comment relates to the fact that I am missing some summary (conclusion) chapter that will gather all this information into a general 1-2 paragraphs. After considering my comments, I think that the article will be complete and suitable for publication in the Microorganisms journal.
Author Response
Reply to the comments received from n.3 anonymous Reviewers for the manuscript by Dell’Anno et alii, entitled “Bacteria, fungi and microalgae for the bioremediation of ma-rine sediments contaminated by petroleum hydrocarbons in the -omics era”, submitted for consideration to Microorganisms.
REVIEWER#1: Thank you for the opportunity to review the article entitled Bacteria, fungi and microalgae for the bioremediation of marine sediments contaminated by petroleum hydrocarbons in the omics era. The article is very interesting and will probably be a very valuable literature item. The authors made a complete review of bioremediation of contaminated marine sediments. In this review, they referred to 201 literature references. In my opinion, the article is written very well in terms of the content and I have no comments of this nature. However, the article needs to be improved in terms of editing. First of all, the authors have to correct the way of editing the text - extract paragraphs; arrange unnecessary spaces and interlining between the text; change the presentation of information in the tables according to journal template. Moreover, after reading the article, I find that the Authors should consider two things in it. First of them is to add some more figures or schemes that refer to the quoted references. This comment is due to the fact that there are many pages in this article that are completely filled with text only. This form is not friendly to the potential reader and does not encourage him to read further. Adding more figures can significantly improve the reader's perception of this article. My second comment relates to the fact that I am missing some summary (conclusion) chapter that will gather all this information into a general 1-2 paragraphs. After considering my comments, I think that the article will be complete and suitable for publication in the Microorganisms journal.
REPLY: We greatly thank REV#1 for her/his appreciations on the value and interest of our work and for the comments for improvement. In the amended version of our manuscript, we have taken in full consideration the suggestions received. Briefly, we i) edited all text and tables to the appropriate fit, ii) we added a novel figure to make the article more accessible and friendly to the readers, and iii) we added the requested conclusions section. We again thank REV#1 for the time spent to help us improve our manuscript.
Reviewer 2 Report
The paper is the result of particularly intense effort and a good capacity for analysis and synthesis that deserves all praise. The importance of using metagenomics and other modern techniques (transcriptomics and metatranscriptomics) to identify new genes and metabolic pathways active in hydrocarbon degradation is also emphasized.
The paper is well written and the concepts are supported by a vast number of references (over 200). The paper is an important up-to-date contribution to the progress of knowledge related to the bioremediation of hydrocarbons in the marine environment.
Author Response
Reply to the comments received from n.3 anonymous Reviewers for the manuscript by Dell’Anno et alii, entitled “Bacteria, fungi and microalgae for the bioremediation of ma-rine sediments contaminated by petroleum hydrocarbons in the -omics era”, submitted for consideration to Microorganisms.
REVIEWER#2: The paper is the result of particularly intense effort and a good capacity for analysis and synthesis that deserves all praise. The importance of using metagenomics and other modern techniques (transcriptomics and metatranscriptomics) to identify new genes and metabolic pathways active in hydrocarbon degradation is also emphasized. The paper is well written and the concepts are supported by a vast number of references (over 200). The paper is an important up-to-date contribution to the progress of knowledge related to the bioremediation of hydrocarbons in the marine environment.
REPLY: As REV#2 did not request any revisions, we would just like to thank REV#2 for the very positive feedback on our manuscript.
Reviewer 3 Report
The manuscript focused on review the current knowledge on the main microbial taxa involved in bioremediation, includingthe main factors influencing bioremediation performance of marine sediments contaminated by petroleum hydrocarbons, as well as the suitability of innovative molecular approaches, which can be useful to enhance the reclamation efficiency of contaminated marine sediments.
1. One of the factors limiting the biodegradation of hydrocarbons is their poor bioavailability due to their hydrophobic nature. It would be good to mention how to increase the bioavailability of hydrocarbons for microbial cells.
2. Table 2 - column "specificity of hydrocarbons" and table 3 "specificity of the compound" - all names should be uniformly written, e.g. in small letters - if they are not proper name.
3. Tabel 2 column "degradaing bacteria" - species names should be written in italics, e.g. aeruginosa, antarctica e.t.c.
4. Line 214 - Staphylococcus - should be written in italics
Author Response
Reply to the comments received from n.3 anonymous Reviewers for the manuscript by Dell’Anno et alii, entitled “Bacteria, fungi and microalgae for the bioremediation of ma-rine sediments contaminated by petroleum hydrocarbons in the -omics era”, submitted for consideration to Microorganisms.
REVIEWER#3
The manuscript focused on review the current knowledge on the main microbial taxa involved in bioremediation, includingthe main factors influencing bioremediation performance of marine sediments contaminated by petroleum hydrocarbons, as well as the suitability of innovative molecular approaches, which can be useful to enhance the reclamation efficiency of contaminated marine sediments.
1. One of the factors limiting the biodegradation of hydrocarbons is their poor bioavailability due to their hydrophobic nature. It would be good to mention how to increase the bioavailability of hydrocarbons for microbial cells.
2. Table 2 - column "specificity of hydrocarbons" and table 3 "specificity of the compound" - all names should be uniformly written, e.g. in small letters - if they are not proper name.
3. Tabel 2 column "degradaing bacteria" - species names should be written in italics, e.g. aeruginosa, antarctica e.t.c.
4. Line 214 - Staphylococcus - should be written in italics
REPLY: We greatly thank REV#3 for her/his minor revisions. These have been taken into account and fully accomplished in the revised version of our manuscript.